# *Leishmania* Mitochondrial Genomes: Maxicircle Structure and Heterogeneity of Minicircles

**DOI:** 10.3390/genes10100758

**Published:** 2019-09-26

**Authors:** Esther Camacho, Alberto Rastrojo, África Sanchiz, Sandra González-de la Fuente, Begoña Aguado, Jose M. Requena

**Affiliations:** Centro de Biología Molecular “Severo Ochoa” (CSIC-UAM), Campus de Excelencia Internacional (CEI) UAM+CSIC, Universidad Autónoma de Madrid, 28049 Madrid, Spain; ecamacho@cbm.csic.es (E.C.); arastrojo@cbm.csic.es (A.R.); asanchiz@cbm.csic.es (Á.S.); sandra.g@cbm.csic.es (S.G.-d.l.F.); baguado@cbm.csic.es (B.A.)

**Keywords:** next-generation sequencing (NGS), de novo assembly, phylogeny, trypanosomatids

## Abstract

The mitochondrial DNA (mtDNA), which is present in almost all eukaryotic organisms, is a useful marker for phylogenetic studies due to its relative high conservation and its inheritance manner. In *Leishmania* and other trypanosomatids, the mtDNA (also referred to as kinetoplast DNA or kDNA) is composed of thousands of minicircles and a few maxicircles, catenated together into a complex network. Maxicircles are functionally similar to other eukaryotic mtDNAs, whereas minicircles are involved in RNA editing of some maxicircle-encoded transcripts. Next-generation sequencing (NGS) is increasingly used for assembling nuclear genomes and, currently, a large number of genomic sequences are available. However, most of the time, the mitochondrial genome is ignored in the genome assembly processes. The aim of this study was to develop a pipeline to assemble *Leishmania* minicircles and maxicircle DNA molecules, exploiting the raw data generated in the NGS projects. As a result, the maxicircle molecules and the plethora of minicircle classes for *Leishmania major*, *Leishmania infantum* and *Leishmania braziliensis* have been characterized. We have observed that whereas the heterogeneity of minicircle sequences existing in a single cell hampers their use for *Leishmania* typing and classification, maxicircles emerge as an extremely robust genetic marker for taxonomic studies within the clade of kinetoplastids.

## 1. Introduction

The genus *Leishmania*, classified within the order Trypanosomatida, belongs to the lineage Excavata, one of the earliest branches diverging from the eukaryotic tree [1]. The trypanosomatids are one of the most successful groups of parasites [2], including organisms of tremendous medical and economic importance, like *Trypanosoma* ssp., causative agents of Chagas disease and sleeping sickness in humans, and *Leishmania* spp., which cause several types of leishmaniasis [3]. Another distinctive feature shared by the trypanosomatids is the presence of a huge DNA network located inside the mitochondrion that is known as kinetoplast DNA (or kDNA). This DNA network consists of dozens of relaxed maxicircles with species-specific sizes, ranging from 20 to 40 kb, and thousands of minicircles ranging from 0.5 to 2 kb [4]. DNA maxicircles are equivalent to the mitochondrial genomes present in the more conventional eukaryotes and they encode mitochondrial rRNAs and several subunits of the electron transport chain complexes. However, many maxicircle transcripts require to be edited before being translated; this post-transcriptional process (known as RNA editing) requires guide RNAs (gRNAs), which are encoded mostly in the minicircles but some are encoded in the maxicircle [5]. The gRNAs direct the proper insertion or deletion of uridine (U) residues to create functional open reading frames in the maxicircle-derived transcripts [4,6]. Nevertheless, the machinery required for trypanosome RNA editing is really complex, comprising several structural proteins and the required enzymes to catalyze cycles of U insertion and deletion [7,8].

In order to completely edit the maxicircle transcripts, a large number of different minicircles are needed for providing the complete set of gRNAs. If one class of minicircles is lost, the parasite cannot complete the editing of the corresponding maxicircle transcripts and therefore it may die. To minimize fatal losses, each class of minicircles is present in several copies and redundant gRNAs are also observed [5]. Moreover, minicircles are replicated in an ordered manner, each DNA molecule is released from the concatenated network and replicated; thereafter, each daughter molecule would be attached to opposite regions of the kDNA network [9]. Once the network doubles in size, it is split in two halves, and these virtually identical networks segregate into daughter mitochondria (and cells).

*Leishmania tarentolae* has been widely used as a model organism for studying RNA editing, and, as a consequence, the structure of its mitochondrial genome (maxicircle and minicircles) is the best characterized to date among the trypanosomatids. Approximately, there are 5000–10000 minicircles and 20–50 maxicircles within the sole mitochondrion existing in this parasite [10]. A 21 kb-fragment, out of the approximately 30 kb in length having the maxicircle molecule of *L. tarentolae*, was sequenced several years ago [11]. The identified genes in the *L. tarentolae* maxicircle [5] encode two small mitochondrial rRNAs (9S and 12S), several proteins of the electron transport chain, including cytochrome b (CYb), cytochrome oxidase subunits I, II and III (COI, COII and COIII), and NADH dehydrogenase subunits 1, 2, 3, 4, 5, 7, 8 and 9 (ND subunits), a ribosomal protein (RpS12) and four open reading frames (ORFs) of unknown function (MURF2, MURF5, G-rich 3 (G3) and G4). Recently, MURF5 has been identified as a component of the mitoribosome and renamed as uS3m [12], and the G3- and G4-encoded proteins have been proposed as subunits of the respiratory complex I [13]. Five of the protein-coding transcripts do not require edition before translation (ND1, ND2, ND4, ND5 and COI), but the transcripts derived from the other 12 protein-coding genes (also known as cryptogenes) need to be edited (and some of them highly edited, or pan-edited) to create translatable mRNAs [14]. A large portion of the *L. tarentolae* maxicircle (around 12 kb) consists of tandem repeats of varying complexities [15]. This region is named the ‘divergent region’ or DR, which is considered as non-coding, even though its precise sequence remains to be elucidated, despite the important efforts done [16].

Apart from *L. tarentolae*, a parasite for the lizard *Tarentola mauritanica*, the coding region of the maxicircle has not been completely sequenced for other *Leishmania* species until very recently (see below). Regarding maxicircles from human pathogenic *Leishmania* species, in 2008, the sequence of an 8.4 kb fragment, belonging to the coding region of the *Leishmania major* maxicircle, was reported [17]. Soon after, most of the coding region for the *L. donovani* (LdBob strain) maxicircle was determined; a notable particularity found in the sequenced maxicircle was a full-size minicircle integrated in the ND1 gene [18]. For *Leishmania mexicana amazonensis* (LV78 strain), though the complete set of mitochondrial pan-edited mRNAs has been characterized, the maxicircle sequence assembly has not been reported [19]. Two fragments of 6535 and 4257 nucleotides belonging to the *Leishmania braziliensis* maxicircle were assembled [20], showing that despite the evolutionary divergence of *L. braziliensis* regarding other pathogenic *Leishmania* species [21], a remarkable conservation in the maxicircle gene ordering (synteny) was maintained. Now, when NGS technology is widely used for determining the nuclear genomic sequences of an increasing number of *Leishmania* species and strains [22,23,24,25,26,27,28,29,30,31,32], the time is coming to generate complete sequences of the mitochondrial genomes for the many *Leishmania* species named to date. In this work, we describe a pipeline designed to assemble the complete maxicircle sequences and the different minicircles by using the genomic NGS-reads generated during *Leishmania* nuclear genome sequencing. For this objective, three of the more pathogenic *Leishmania* species were selected: *Leishmania infantum,* causative agent of visceral leishmaniasis in the Mediterranean Basin, the Middle East, central Asia, South America and Central America, *L. braziliensis*, which causes metastasizing mucocutaneous leishmaniasis in South America, and *L. major*, responsible for cutaneous leishmaniasis in Iran, Saudi Arabia, North Africa, the Middle East, Central Asia and West Africa [33]. As a result, we are reporting the sequences of the complete coding-region of the maxicircles from *L. major* (Friedlin strain), *L. infantum* (JPCM5 strain) and *L. braziliensis* (M2904 strain), and many of the distinct minicircles existing in these species. Finally, we assessed the usefulness of both types of mitochondrial sequences for phylogenetic analyses. A robust phylogenetic tree was constructed based on the complete coding-region from the relatively few maxicircles to date sequenced and characterized in kinetoplastids.

## 2. Materials and Methods

### 2.1. Parasites, DNA Isolation and Illumina Sequencing 

Promastigote cultures of *L. major* (MHOM/IL/81/Friedlin), *L. infantum* (MCAN/ES/98/LLM-724; JPCM5) and *L. braziliensis* (MHOM/BR/75/M2904) were provided by Dr. J. Moreno (WHO Collaborating Centre for Leishmaniasis, Centro Nacional de Microbiología, Instituto de Salud Carlos IIII, Madrid, Spain). Promastigotes were cultured at 26 °C in M199 medium (Sigma-Aldrich), supplemented with 10% heat-inactivated fetal calf serum. Genomic DNA was isolated following the classical phenol-chloroform-isoamyl alcohol extraction method as described elsewhere [34].

Construction of libraries from total DNA and paired-end library sequencing were performed at the Centro Nacional de Análisis Genómico (CNAG-CRG, Spain) using Illumina HiSeq 2000 technology. Reads having unexpected combinations of indices (due to barcode swapping or index hopping) were removed from the data. The number of 2 × 125 nucleotide (nt) reads generated for each *Leishmania* species were: 53,340,123 for *L. major*; 56,833,213 for *L. infantum* and 51,050,312 for *L. braziliensis*. Further information about the nuclear genome sequencing and assembly for these three species is available at http://leish-esp.cbm.uam.es/ and in previous publications [30,32].

### 2.2. Assembly of Leishmania Maxicircle and Minicircle Sequences From Total DNA NGS Short Reads

Illumina reads were aligned by Bowtie2 [35] against the *Leishmania* genome sequences available at Leish-ESP website (http://leish-esp.cbm.uam.es), TriTrypDB (https://tritrypdb.org), NCBI (https://www.ncbi.nlm.nih.gov) and EMBL-EBI (https://www.ebi.ac.uk/). The SAM files were converted into BAM files using Samtools (http://www.htslib.org/) and the unaligned reads were extracted using bedtools2 [36]. PrinseqQuality (http://prinseq.sourceforge.net/) was applied to quality filtering/trimming of the unaligned reads (cut-off value, 20), and only reads with length ≥60 nt were used. The exact parameters were: -trim_qual_right 25, -trim_qual_left 25, -trim_qual_type mean, -trim_qual_window 5, -trim_qual_step 1, -trim_ns_left 1, -trim_ns_right 1, -ns_max_p 1, -ns_max_n 3, -min_qual_mean 25 and -min_len 60. Pair reads were selected and ordered using in-house scripts. These reads were used to generate assemblies (contigs) by the IDBA_UD (version 1.1.1) assembler [37], selecting parameters: --mink 20, --maxk 120, --min_support 1, --min_contig 500 and --pre_correction. Contigs with sizes above 500 nucleotides were analyzed by BLAST searches at NCBI; BLAST results showing E-values lower than 0.01 and scores larger than 35 were selected. Finally, the contigs sharing sequence homology with entries corresponding to maxicircles and minicircles sequences of trypanosomatids were selected for further analyses.

Putative minicircle contigs were further tested for circularity, manifested by the presence of direct repeated ends in the assembled molecule. Briefly, contigs were split into two halves, and then Minimus2 (from toAmos (version 3.1.0)) tool was used to identify repeated ends as described elsewhere [38]. Finally, those contigs with direct repeated ends from 40 nt up to 1.5 times the kmer size (120 nt) were considered as circular.

The complete pipeline, as well as the associated in-house scripts, have been deposited in GitHub (https://github.com/genomics-ngsCBMSO/Mithocondrial-assembly.git).

### 2.3. Validation of Assemblies by PCR Amplification

Experimental validation of the precise overlapping between the two contigs (LmjF_max-CgL and LmjF_max-CgS) containing *L. major* maxicircle sequences was performed by PCR, using the following two oligonucleotides: LmjF-Mxc-2, 5′-CCTGCCCAAT GATTGTATGA-3′ and LmjF-Mxc-4, 5′-CCATACCGAG AGTATCAATCTTG-3′. Similarly, to determine the accuracy of *L. major* assembled minicircles, two primers (forward, 5′-AATCGAAAAA TGGGTGCAGA AATCC-3′, and reverse, 5′-GATTTTCGCA GAACGCCCCT AC-3′) were designed within the conserved region of LmjF-mic1; both oligonucleotides overlap each other by 4 nt at the 5′-end. Amplification was carried out in a 50 µL-final volume using the Phusion Hot-Start II DNA polymerase (ThermoFisher Scientific) and 100 ng of total DNA; PCR was run on a Techne TC-3000 PCR Thermal Cycler using the following profile: Initial denaturation at 98 °C for 30 s followed by 35 cycles of 10 s at 98 °C, 30 s at 66.5 °C and 45 s at 72 °C, completed by a final incubation of 5 min at 72 °C. The amplification products were purified using the Favorgen FavorPrep GEL/PCR Purification kit and cloned into pNZY28 vector using the NZY-blunt PCR cloning kit (Nzytech). After transformation of *Escherichia coli* DH5α bacteria and antibiotic selection, plasmid DNA from bacteria cultures was extracted using the AccuPrep Plasmid MiniExtraction kit (Biomedal). Clones were checked by restriction endonuclease analysis and sequenced at the Genomic Unit from Universidad Complutense de Madrid. The clone sequences were edited using Bioedit Software (Ibis Biosciences, Carlsbad, CA, USA) and then compared to the in-silico assembled sequences.

### 2.4. Phylogenetic Analysis

Phylogenetic trees were constructed using either the full sequence of minicircles or the entire coding region of the maxicircles. The sequences were aligned using Clustal Omega (https://www.ebi.ac.uk/Tools/msa/clustalo/). Manual refinements were omitted. Phylogenetic relationships were inferred by using the maximum likelihood method and Tamura-Nei model [39]. Initial trees for the heuristic search were obtained automatically by applying neighbor-join (NJ) and BioNJ algorithms to a matrix of pairwise distances estimated using the maximum composite likelihood (MCL) approach, and then selecting the topology with superior log likelihood value. Bootstrap values were derived from 500 replicates. For the phylogenetic analysis of minicircles, the option of non-coding DNA was selected, whereas for maxicircles phylogeny, the code for plant plastids was selected. All the evolutionary analyses were conducted in MEGA X [40].

### 2.5. Analysis of Conserved Motifs in Leishmania Minicircles by LOGOS Software

The conserved regions of minicircles for each one of the species, *L. major*, *L. infantum* and *L. braziliensis*, were analyzed using the WebLogo 3 (version 3.7.4) tool [41]. For the output files, the classic option was selected.

### 2.6. Data Availability

The sequencing NGS-reads (used for the assemblies reported in this work) and the assembled sequences have been submitted to the European Nucleotide Archive (ENA) and are available under the study accession number PRJEB33887. The ENA/GenBank accession numbers of the assembled sequences are: LR697134 for the *L. braziliensis* maxicircle, LR697136 for the *L. adleri* maxicircle, LR697135 for *L. guyanensis* maxicircle, LR697137 for *L. infantum* maxicircle and LR697138 for the *L. major* maxicircle. Additionally, the sequences for the *L. major*, *L. infantum* and *L. braziliensis* maxicircles and minicircles can be downloaded at the Website Leish-ESP (http://leish-esp.cbm.uam.es).

## 3. Results 

### 3.1. Assembly and Annotation of the L. major Maxicircle

The initial question of this work was whether the assembly of the mitochondrial genome (i.e., maxicircle) would be possible using the sequence reads generated by the Illumina platform in a *Leishmania* whole-DNA sequencing experiment. For this purpose, we used the reads derived from an on-going project aimed to the de novo assembly of the *L. major* (Friedlin strain) nuclear genome (manuscript in preparation). Based on a strategy previously used for uncovering seven misassembled regions in the *L. major* nuclear genome using RNA-seq reads [42], we designed the pipeline outlined in Figure 1 and detailed in the Methods section. Firstly, the 53,340,123 paired-end reads, generated from the *L. major* whole-DNA sequencing, were mapped to the reference genome (version 2016, available at http://leish-esp.cbm.uam.es/L_major_downloads.html). Above four million failed to align and, after further filtering by quality, 3,793,390 pair-end reads were selected to carry out a de novo assembly. As a result, 153 contigs longer than 500 nt were obtained. Two of them showed sequence identity with maxicircle entries found in the GenBank/ENA databases. The longest one, named LmjF_max-CgL (17273 nt in length) was found to include the 8738 nt maxicircle sequence described by Yatawara et al [17], sharing a sequence identity higher than 99%. The second contig, named LmjF_max-CgS, was 2230 nt in length. A BLAST-alignment between contigs LmjF_max-CgL and LmjF_max-CgS suggested a possible overlapping between the extremities of both sequences. This was experimentally validated by PCR amplification, cloning and sequencing (see Methods), and, as a result, a continuous contig of 18998 nt in length was assembled. The extremities of this contig, named LmjF_max-CgT, consisted of repeated A + T rich sequences; the attempts to PCR amplify the region separating the LmjF_max-CgT sequence extremities were unfruitful. Nevertheless, these results were not surprising as amplification of the maxicircle region, known as divergent region (DR), has been attained only partially in spite of the important efforts done [43,44]. To the best of our knowledge, only for the *Trypanosoma brucei* EATRO strain 427 the maxicircle sequence has been completed, yielding a molecule of 23016 bp in length [45]. According to a Southern blot analysis, the *L. major* (Friedlin) maxicircle would have a size around 31 kb in length.

BLAST alignments against the *L. tarentolae* maxicircle (GenBank accession number M10126.1) showed that the coding region of the assembled *L. major* maxicircle was highly conserved in both sequence and gene order. Table 1 shows the location and conservation of the genes identified in the *L. major* maxicircle. Figure 2A shows a physical map of the *L. major* assembled sequence and the reads coverage. The depth-coverage of reads on the region located upstream of the coding region in the maxicircle contig evidenced the low complexity and repetitive nature of the DR region.

### 3.2. Assembly and Annotation of the Maxicircles in L. infantum, L. braziliensis and Other Leishmania Species

The success achieved in the assembly of the *L. major* maxicircle prompted us to repeat the strategy for two other *Leishmania* species, *L. infantum* and *L. braziliensis*, whose genome assemblies have been recently improved by our group [30,32]. Following the pipeline depicted in Figure 1, Illumina reads derived from the sequencing of total DNA were mapped to the reference genomes of these two *Leishmania* species (TriTrypDB.org; leish-esp.cbm.uam.es). After mapping and filtering, 4,057,401 and 1,347,958 paired-end reads for *L. infantum* and *L. braziliensis*, respectively, remained as non-aligned reads. These reads were used for the de novo assembly into contigs. For each species, a large contig and several small ones were generated. The large contigs were found to contain all the maxicircle genes (Figure 2B,C) and their sizes were 17897 nt for the *L. infantum* longest-contig and 17771 nt for the *L. braziliensis* one. Again, as experienced during the *L. major* maxicircle assembly, the DR region could not be entirely assembled, due to its highly repeated structure. Table 1 contains the coordinates for all genes present in the maxicircles of these species. In spite of the evolutionary distance existing between these *Leishmania* species, a remarkable conservation, both in structure and gene sequence, was evidenced among the maxicircles in these three *Leishmania* species and also with the *L. tarentolae* maxicircle, which was used as a reference for gene annotation [5]. Moreover, this high degree of conservation in the gene order is also maintained when compared with the maxicircle structure in species of the genus *Trypanosoma* [46], a fact that results astonishing considering that the common ancestor of *Leishmania* and *Trypanosoma* genera existed hundreds of millions years ago [47].

In summary, we have demonstrated that it was possible to determine the complete coding region of the maxicircles for *L. major*, *L. infantum* and *L. braziliensis* using Illumina reads derived from whole-DNA sequencing (Figure 2). Moreover, the quality of the mitochondrial genomes generated was well supported by the high coverage of reads used to build the assemblies. Thus, the mean coverage values (excluding the divergent region) were 1241×, 3319× and 929× in the maxicircle sequences of *L. major*, *L. infantum* and *L. braziliensis*, respectively.

Moreover, as an additional proof of concept, we used Illumina reads produced by other laboratories and undertook the task of assembling the mitochondrial genome, following the pipeline shown in Figure 1. Thus, firstly, we accessed the Illumina raw reads generated in the sequencing of total DNA from *Leishmania adleri*, a parasite belonging to the lizard-infecting Sauroleishmania subgenus [48], and used them for de novo assembling. As a result, we assembled a 19219 nt long contig, which expands the entire gene-coding region. In a similar way, using the Illumina raw reads generated in the genome sequencing project for *Leishmania guyanensis* [49], a 19624 bp long contig that completely covers the maxicircle coding-region was obtained for this species of the Viannia subgenus.

### 3.3. Assembly of Minicircle Sequences and Analysis of the Minicircle Populations Existing in L. major, L. infantum and L. braziliensis

In addition to the maxicircles, the de novo assembly yielded a large number of smaller contigs in all three species. BLAST searches confirmed that most of them contained minicircle sequences. Furthermore, we found that a large fraction of these sequences would represent complete minicircles, since they could be assembled as circular molecules. Table 2 summarizes the number of different minicircles identified in the three *Leishmania* species analyzed. Within a given species, the sequence identity between two different minicircles was found to be 95% or lower.

Although minicircles characterized in species of genera *Crithidia, Leishmania* and *Trypanosoma* were heterogeneous in size and sequence, a common feature was the presence of a conserved region, ranging from 100 to 200 nt [50]. Within this region, a 12 nt conserved sequence, designated universal minicircle sequence (UMS or CSB-3, 5′GGGGTTGGTGTA-3′), early emerged as the characteristic signature of this class of molecules. Furthermore, two additional conserved sequence blocks CSB-1 (5′-AgGGGCGTTC-3′) and CSB-2 (5′-cCCCGTNC-3′) were defined [50]. CSB-3 is the central part of the minicircle replication origin, being the binding site for the UMS binding protein (UMSBP), a protein involved in kDNA replication and segregation [51].

The presence of these conserved sequence motifs was analyzed in the minicircles assembled in the three *Leishmania* species. Interestingly, the order and sequence of the CSB boxes were conserved in all *L. major* minicircles. In 77 out of the 92 minicircles assembled in *L. infantum*, these motifs also appeared in the expected order; the other 15 sequences showed deviations either in the order or the absence of some of the CSB boxes. For these reasons, these molecules were categorized as ‘uncertain’ (Table 2), and their real existence needs to be addressed by direct experimental methods. In contrast, when the process was repeated with the *L. braziliensis* data, the number of bona fide assembled minicircles was only three circular molecules and another three linear ones (Table 2). Additionally, 22 of the assembled molecules contained only one or two CSB (uncertain minicircles). We have not found a clear explanation to these relatively poor results obtained in the *L. braziliensis* minicircle assembling process, considering the large number of different minicircles assembled in *L. major* and *L. infantum* species (Table 2). Although some of the differences in the number of assembled minicircles may be due, in part, to the lower number of non-aligned reads at the beginning of the de novo assembly (see above), it cannot be excluded an impoverishment in the minicircle population due to the repeated axenic culturing of this strain.

In recent work, Kocher and co-workers [52] carried out an in-depth analysis of the minicircle populations in several New World *Leishmania* species. For this purpose, these authors amplified by PCR the 120 bp long minicircle conserved region; after high-throughput sequencing of the PCR products, they found significant numbers of distinct minicircle sequences from each strain. Recently, these authors have deposited in the GenBank database, 42 apparently full-size minicircles for *L. braziliensis* M2904 strain (accession numbers KY698780-KY698821). According to an author’s comment accompanying the entries, these sequences were assembled from Illumina reads using the Velvet tool. Interestingly, but not surprisingly as both studies have been done using the same strain, sequence identities higher than 99% exist between LbrM-mic1 and KY698625 sequence, LbrM-mic3 and KY698802, and LbrM-mic5 and KY698812.

Table 2 also shows the size range and mean size of the complete minicircles assembled for the three *Leishmania* species. Although the size range in the *L. major* minicircles was higher than those observed for the *L. infantum* and *L. braziliensis* minicircles, the mean size was clearly shorter in *L. major*, 691 nt versus 797 nt and 744 nt for the *L. infantum* and *L. braziliensis* minicircles, respectively. Furthermore, the values calculated from the 42 *L. braziliensis* (M2904 strain) minicircles deposited in the GenBank database (see above) were similar (size range of 738 to 763 nt and a mean size of 748 nt) to those assembled in this work.

Based on the position of the well-conserved CSB elements, we aligned the minicircles identified in these three *Leishmania* species. For this study, the 42 sequences for *L. braziliensis* minicircles available at GenBank (see above) were used. The results, shown in Figure 3, evidenced the existence of a conserved region extending 150–152 nucleotides, in which all three CSB elements were present at equivalent positions in the consensus sequence. Remarkably, in the *L. braziliensis* minicircles, a second CSB-2-like motif (CCCCGTGC) was found to be well conserved (Figure 3C); nevertheless, this CSB-2-like motif was not conserved in the *L. major* and *L. infantum* minicircles. Overall, and expectedly, the conserved regions of *L. major* and *L. infantum* minicircles were more similar to each other than to *L. braziliensis* minicircles.

### 3.4. Experimental Validation of In-Silico Assembled Minicircles

Although all the assembled circular molecules and most of the linear ones contained the three conserved motifs (CBS-1, -2 and -3), a PCR-based experiment was conducted to validate the correctness of the *L. major* assembled minicircles. As the oligonucleotides were designed within the conserved region of LmjF-mic1 minicircle (see Figure 4A), it was expected to amplify several different minicircles; indeed, the sequence of these oligonucleotides was found 100% identical in minicircles LmjF-mic8, LmjF-mic39, LmjF-mic63 and LmjF-mic92, and in many others with a substantial sequence identity. After PCR amplification, cloning and enzyme restriction analysis (Figure 4B), four recombinant clones (numbers 2, 6, 7 and 9) were sequenced. Each sequence matched with some of the in silico assembled minicircles: Clone 2 showed 99.6% sequence identity with LmjF-mic1, clone 6 was 98% identical to LmjF-mic98 and clones 7 and 9 were 99.7% identical to LmjF-mic87 and LmjF-mic28, respectively. Remarkably, from an in-depth analysis of the NGS-reads covering the polymorphic positions, it was observed that both molecules indeed were co-existing in the minicircle population. Figure 4C illustrates this finding for the pair clone 2/LmjF-mic1; there are two C/T polymorphic sites that were assembled as T in the LmjF-mic1 (due to the slightly higher frequencies of the T residues over the C ones), but they appeared as C residues in clone 2. This finding suggested the existence of micro-heterogeneities in the molecules derived from a given minicircle class, in a similar manner as occurs in viral quasispecies [53]. Micro-heterogeneities or polymorphisms within a minicircle class have been reported previously in *L. tarentolae* [5].

### 3.5. Usefulness of Minicircle and Maxicircles Sequences for Species Identification and Phylogenetic Analysis

Firstly, a phylogenetic tree was generated from the 97 complete minicircles assembled for *L. major* (Figure 5). The picture evidenced that minicircles constitute a heterogeneous family, in which both very similar and highly divergent molecules co-exist. To test whether, in spite of this heterogeneity, minicircle phylogenies would be a robust market for species identification, seven sequences were randomly selected from each one of the minicircles sets assembled for *L. major*, *L. infantum* and *L. braziliensis*, and an evolutionary analysis by the maximum likelihood (ML) method was conducted. As shown in Figure 6A, minicircle sequences of each species did not form species-specific monophyletic clusters, but they were distributed in several groups comprising sequences from different species. Moreover, some of the bootstrap values were very low, indicating that the constructed tree was not robust (reliable). These results suggest that *Leishmania* typing based on sequencing a single minicircle may lead to miss-identification, considering that several distinct minicircle classes coexist in a single parasite.

On the contrary, as stated recently [54], maxicircle-based taxonomy of *Leishmania* and related trypanosomatids might be consistent and reliable. To investigate this possibility further, we inspected GenBank and retrieved those maxicircle sequences that contain the entire gene-coding region. However, among the large number of maxicircle sequences currently deposited in this database, only thirteen were found to cover the gene-coding region completely. These sequences, together with the five maxicircles assembled in this work, were used to construct a phylogenetic tree using the ML method (Figure 6B). The phylogenetic relationships depicted in this analysis showed a total agreement with an equivalent study based on alignments of concatenated gene sequences (i.e., 18S rDNA, gGAPDH, RPOIILS and HSP70) [55]. Moreover, the bootstrap analysis indicated an extremely robust structure of the tree branches, which are supported by very high percentages, showing 100% bootstrap confidence for the separation of all *Leishmania* species included in the study. In summary, these results demonstrated that maxicircle-based phylogenetic analysis may represent a robust method to be employed for *Leishmania* typing and classification, and a reliable method for establishing the taxonomy of trypanosomatids.

## 4. Discussion

Mitochondrial genes and genomes have long been applied in molecular evolution and population genetics. With the development of sequencing technologies, an increasing number of mitochondrial genomes have been sequenced in different groups of organisms [56,57,58]. Paradoxically, until very recently, the mitochondrial genomes of *Leishmania* parasites have been neglected in the gold era of nuclear genome sequencing. In the field, a few works, in which NGS technologies are used to determine *Leishmania* mitochondrial genomes, have been published [5,54,59]. In these studies, either previous kDNA purification [5] or selective PCR-amplification of maxicircles [54] steps were included. We hypothesized that such a specific purification/amplification might be not needed, and the complex mitochondrial genome of these organisms might be determined using the NGS reads generated in the projects aimed to assemble the nuclear genomes. Here, we provided a pipeline illustrating how the complex *Leishmania* mitochondrial genome, composed by a conserved maxicircle and a heterogeneous population of minicircles, could be assembled by sequence reads derived from massive sequencing of total DNA.

Soon after the incorporation of PCR amplification to diagnosis, kDNA minicircles were envisaged as an adequate molecular target for sensitive *Leishmania* detection because they are present in a high copy number, and contain conserved sequence blocks in which primers may be designed to amplify these molecules in a broad spectrum of strains and species [60]. In fact, qPCR assays targeting minicircle DNA have been proved highly sensitive and accurate for the detection and quantification of *Leishmania*, even in lesion biopsy specimens [61].

However, the usefulness of minicircle sequences for phylogenetic analysis is being put into question, given the high level of nucleotide polymorphisms amongst the thousands of copies present in a single cell [62]. In good agreement, our data (Figure 6A) also evidenced that the use of individual minicircle sequences may generate disparate phylogenetic relationships between the *Leishmania* species. Only application of massive sequencing approaches to delineate the minicircle compendium found in a single *Leishmania* strain might allow reliable identifications of *Leishmania* species [52], but this would increase greatly the complexity of the analysis. Nevertheless, at this point, when NGS methodologies are becoming routine techniques, the coding region of the maxicircle, because of its long size (and homogeneity in gene order), emerges as a suitable phylogenetic marker for *Leishmania* typing and, consequently, for exploring evolutionary relationships within the trypanosomatids [54]. In fact, the sequence of the cytochrome b (Cyt b) gene, which is part of the maxicircle, is being used in many studies for phylogeny analysis within the *Leishmania* genus [63,64]. Moreover, the Cyt b gene is included in many of the multilocus sequence typing (MLST) sets currently used for phylogenetic studies in *Leishmania* [65]. MLST is based on the use of concatenated sequences from several genes and is aimed to ensure robust phylogenetic inferences. Nevertheless, this approach needs individual amplification and sequencing of each gene composing the MLST set. In contrast, the coding region of maxicircles represents a natural concatenation of genes, which vary in function and show different evolutionary rates. In fact, as shown in this work and in a recent article [54], phylogenetic analyses based on the maxicircle coding region yield extremely robust trees.

In summary, high-throughput sequencing is becoming a routine methodology for addressing many molecular aspects, from genomic structure to gene expression analysis, in every organism. Typing and phylogenetic analysis could also be improved by the use of genome-wide sequencing data. Here, a pipeline was described and used to assemble the mitochondrial genome (minicircles and maxicircle) from NGS reads generated by sequencing of total DNA *Leishmania*; this strategy would be also suitable for assembling these molecules in related trypanosomatids. Additionally, the usefulness of these molecules for phylogenetic purposes was evaluated. As a result, the entire coding region of the maxicircle sequence emerged as a robust marker for studying the taxonomy of both *Leishmania* and related trypanosomatids.

## Figures and Tables

**Figure 1 genes-10-00758-f001:**
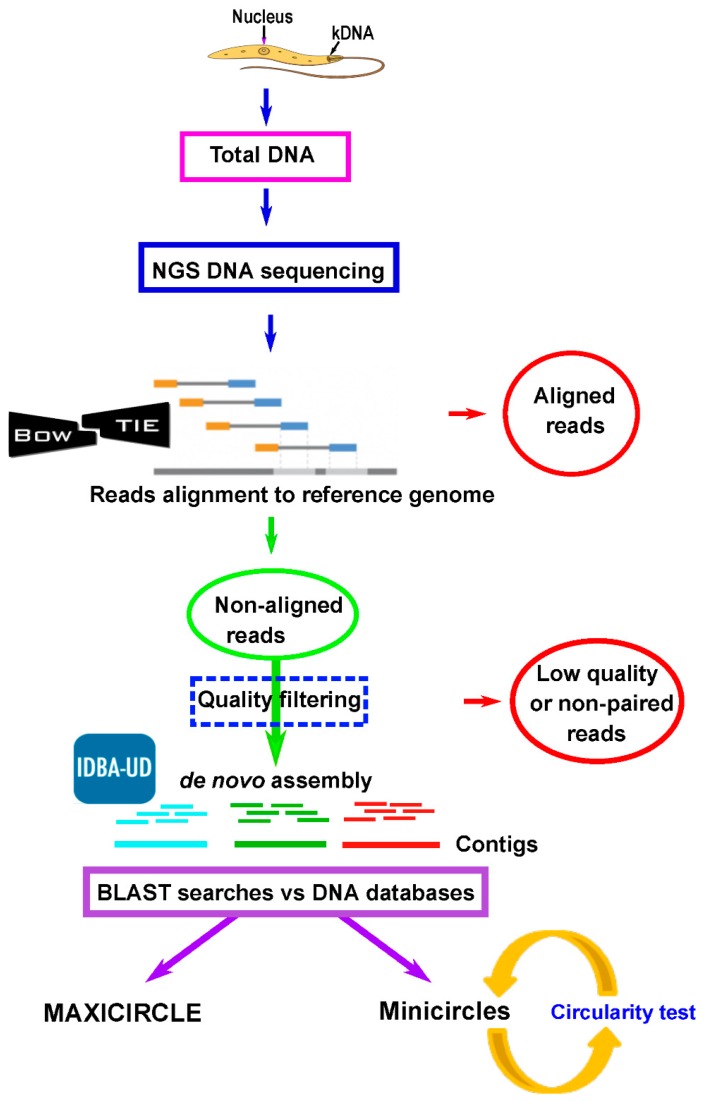
Workflow for the assembly of maxicircle and minicircle molecules from reads obtained by next-generation sequencing (NGS) of *Leishmania* total DNA. An alignment against the reference nuclear genome using Bowtie2 was carried out in order to separate the non-aligned reads. After a quality filtering using Prinseq, a de novo assembly was performed with IDBA-UD, and contigs longer than 500 nt were analyzed using the NCBI-BLAST tool to evaluate sequence identities with reported minicircle and maxicircle sequences. Minicircle sequences were further analyzed by a circularity test.

**Figure 2 genes-10-00758-f002:**
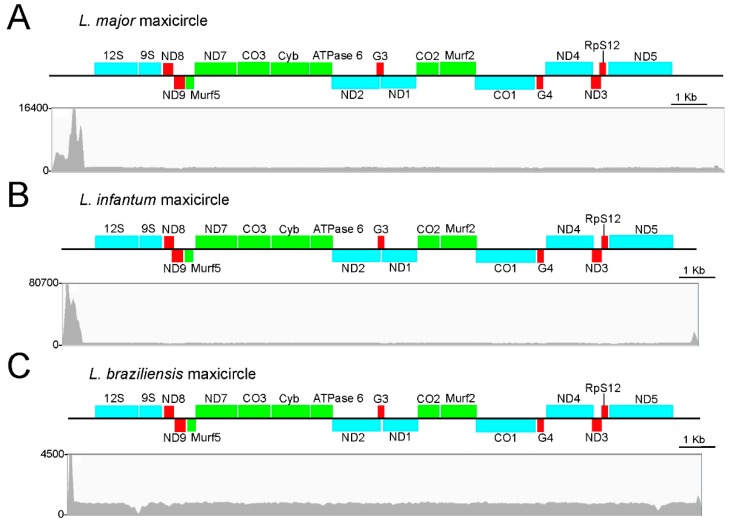
Diagram of the assembled contigs corresponding to the *L. major* (**A**), *L. infantum* (**B**) and *L. braziliensis* (**C**) maxicircles. The nomenclature and abbreviations of genes are those established for *L. tarentolae* [5]. Blue, green and red boxes are used to indicate non-edited genes, edited genes and pan-edited genes, respectively. The graph, below each gene map, shows the read depth, i.e., the number of reads aligned in a given position along the assembled contig sequence (linear scale).

**Figure 3 genes-10-00758-f003:**
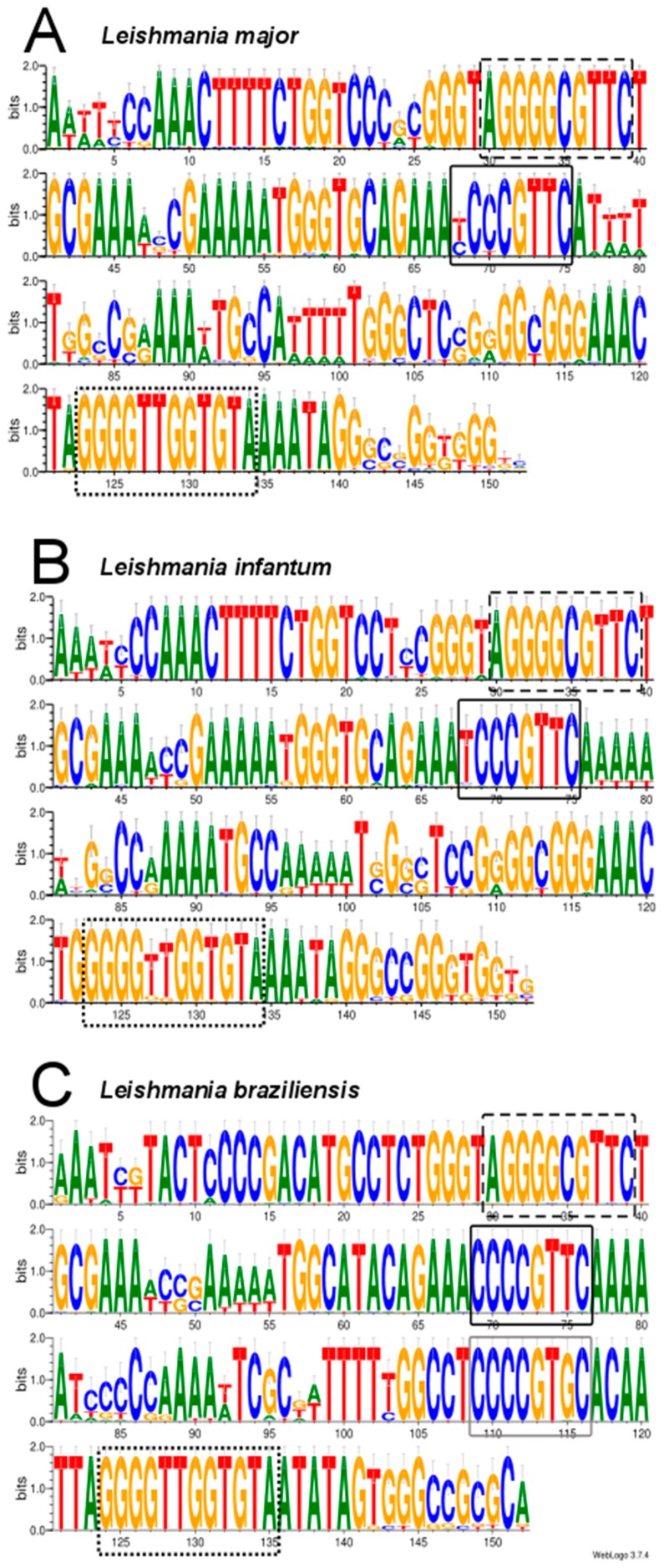
Sequence conservation analysis among the different minicircles assembled for *L. major* (**A**), *L. infantum* (**B**) and *L. braziliensis* (**C**). Images were generated using the WebLogo 3 server [41]. The conserved motifs CSB1 (dashed-black box), CSB2 (black box) and CSB3 (dotted-black box) are highlighted. In the *L. braziliensis* sequence, a CSB-2 like sequence is boxed in grey.

**Figure 4 genes-10-00758-f004:**
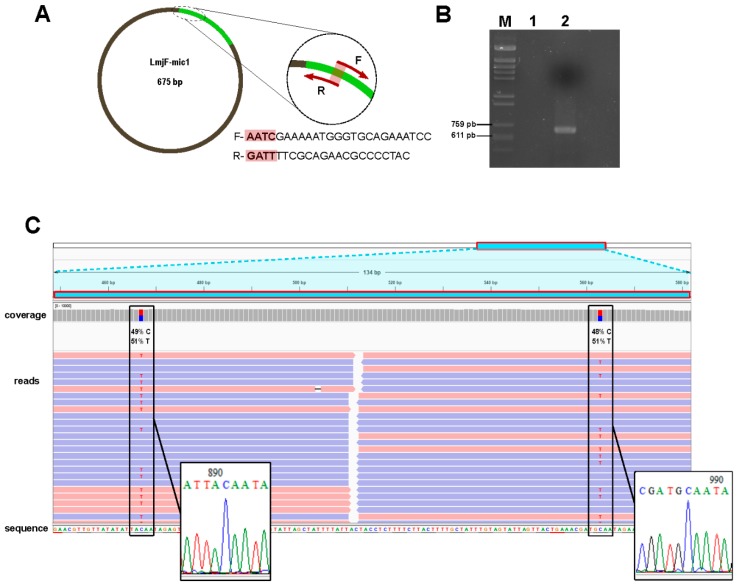
PCR validation of minicircle assemblies. (**A**) Primer design scheme based on the *L. major* minicircle 1 (LmjF-mic1). Primers F (forward) and R (reverse), shown as maroon arrows, were designed on the conserved region of the minicircle (green area). The primers overlap by 4-nt. (**B**) Analysis by agarose gel electrophoresis of the PCR amplification product (lane 2) and negative control (lane 1) contains the amplification in the absence of DNA template. *Hin*dIII-digested phi29-DNA was used as molecular weight marker (lane M) loaded; the length (in bp) of the marker bands with a size close to the amplification product are indicated on the left. (**C**) Alignment of Illumina reads on the sequence of clone 2 (one of the clones derived from the cloning of the amplicon shown in B). Two polymorphic sites (boxes) were identified and the frequency (in percentage) of the nucleotides found at those positions are shown. The insets at the bottom show the chromatogram of the sequence determined by the Sanger method in clone 2.

**Figure 5 genes-10-00758-f005:**
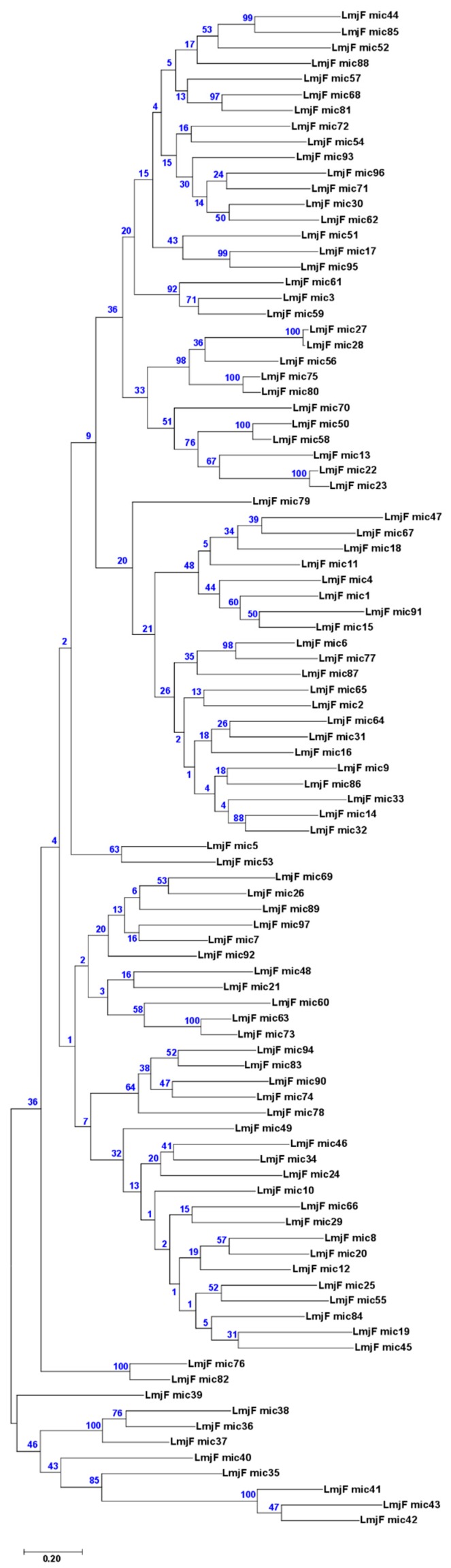
Sequence divergence among the 97 *L. major* assembled minicircles (circular) inferred by evolutionary analysis. The analysis was carried using the maximum likelihood method and Tamura-Nei model [39]. The tree with the highest log likelihood (–67786.96) is shown. The percentage of trees in which the associated taxa clustered together is shown next to the branches (bootstrap values were obtained from 500 replicates). Initial tree(s) for the heuristic search were obtained automatically by applying the neighbor-join (NJ) and BioNJ algorithms to a matrix of pairwise distances estimated using the maximum composite likelihood (MCL) approach, and then selecting the topology with superior log likelihood value. The tree is drawn to scale, with branch lengths measured in the number of substitutions per site. There were 936 positions in the final dataset.

**Figure 6 genes-10-00758-f006:**
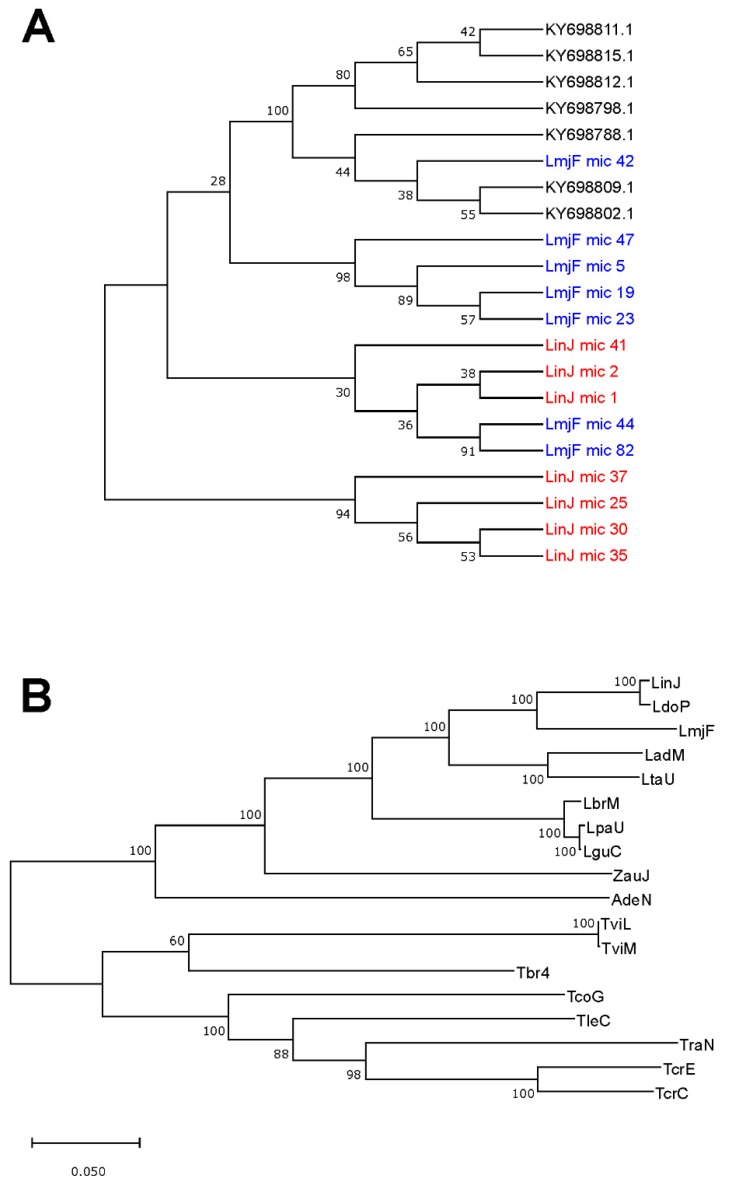
Analysis of the usefulness of minicircle and maxicircle sequences for phylogenetic purposes. (**A**) Phylogenetic tree obtained by sequence comparison of seven minicircles randomly selected for each one of the following species: *L. major* (LmjF mic), *L. infantum* (LinJ mic) and *L. braziliensis* (GenBank accession codes). (**B**) Phylogenetic tree based on the maxicircle sequence of the following trypanosomatids: LinJ, *L. infantum* (this work); Ldo, *L. donovani* (GB: CP022652.1); LmjF, *L. major* (this work); LadM, *L. adleri* (this work); LtaU, *L. tarentolae* (GB: M10126.1); LbrM, *L. braziliensis* (this work); LpaU, *L. panamensis* (GB: MK570510.1); LguC, *L. guyanensis* (this work); ZauJ, *Zelonia australiensis* (GB: MK514117); AdeN, *Angomonas deanei* (GB: KJ778684.1); TviL (GB: KM386509.1) and TviM (GB: KM386508.1), *Trypanosoma vivax*; Tbr4 (GB: M94286.1), *T. brucei*; TcoG (GB: MG948557.1), *T. copemani*; TleC (GB: KR072974.1), *T. lewisi*; TraN (GB: KJ803830.1), *T. rangeli*; TcrE (GB: DQ343646.1) and TcrC (GB: DQ343645.1), *T. cruzi*. The phylogenetic analyses were carried out as indicated in the legend of Figure 5. The log likelihood of the trees shown in A and B was –16227.07 and –143206.12, respectively. The percentage of trees in which the associated taxa clustered together is shown next to the branches (bootstrap values were obtained from 500 replicates).

**Table 1 genes-10-00758-t001:** Genes and their location in the *Leishmania major*, *L. infantum* and *L. braziliensis* maxicircles.

	Coordinates (strand: Plus (+) or minus (−))		
Gene ^a^	*L. major*	*L. infantum*	*L. braziliensis*
12S rRNA	1302-2462 (+)	993-2156 (+)	815-1974 (+)
9S rRNA	2498-3107 (+)	2181-2790 (+)	2005-2615 (+)
ND8	3200-3450 (+)	2885-3198 (+)	2688-2878 (+)
ND9	3511-3802 (−)	3185-3524 (−)	3086-3313 (−)
MURF5	3777-4082 (−)	3496-3792 (−)	3333-3632 (−)
ND7	4095-5271 (+)	3806-4979 (+)	3650-4802 (+)
CO3	5275-6155 (+)	4999-5879 (+)	4884-5719 (+)
CYb	6166-7270 (+)	5889-6998 (+)	5743-6830 (+)
ATPase 6	7336-7934 (+)	7061-7660 (+)	6878-7467 (+)
ND2	7939-9272 (−)	7665-8996 (−)	7482-8807 (−)
G3	9232-9397 (+)	8958-9120 (+)	8775-8941 (+)
ND1	9361-10302 (−)	9085-10026 (−)	8930-9855 (−)
CO2	10311-10939 (+)	10029-10657 (+)	9862-10490 (+)
MURF2	10931-12009 (+)	10649-11728 (+)	10482-11573 (+)
CO1	12000-13649 (−)	11719-13368 (−)	11564-13196 (−)
G4	13698-13861 (−)	13372-13493 (−)	13292-13459 (−)
ND4	13987-15299 (+)	13705-15017 (+)	13512-14823 (+)
G5 (ND3)	15282-15524 (−)	15000-15218 (−)	14806-15004 (−)
RPS12	15474-15670 (+)	15258-15425 (+)	15036-15191 (+)
ND5	15751-17522 (+)	15493-17265 (+)	15253-17024 (+)

^a^ The names for individual genes are those used for *L. tarentolae* mitochondrial genes [5].

**Table 2 genes-10-00758-t002:** Minicircles assembled in *L. major*, *L. infantum* and *L. braziliensis.*

Minicircle Type	Circular			Linear/Partial	Uncertain	Total
Species	Number	Size Range	Mean Size			
*L. major* (Friedlin)	97	660–876	691	7	-	104
*L. infantum* (JPCM5)	49	775–832	797	28	15	92
*L. braziliensis* (M2904)	3	741–749	744	3	22	28

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
