# Peer review of "Leishmania Mitochondrial Genomes: Maxicircle Structure and Heterogeneity of Minicircles"

_genes, 2019, doi:10.3390/genes10100758_

Round 1

Reviewer 1 Report

In this study, the authors developed a pipeline to assemble mini and maxicircle mitochondrial genome structures from raw genomic reads of several Leishmania species. The authors then combined their newly assembled mt genomes with existing mt genome data to assess the phylogenetic utility of mini and maxicircles. They found that maxicircles are phylogenetically informative, whereas minicircles have considerable heterogeneity within a single cell and are thus not ideal as phylogenetic markers. The Leishmania mt genome network seems like an ideal system to develop assembly pipelines and assess the phylogenetic utility of variable mt genome structures. I think the bioinformatic approach is creative and could be applicable to other systems (although the authors do not discuss this point). Their results also have important implications for future phylogenetic work in trypanosomatids.

However, I do have several questions and concerns that the authors should address.

1) There are parts of the introduction that can be either condensed or expanded. To condense, I suggest combining the background information on kDNA and maxi/minicircles from the first three paragraphs. At the same time, I think you should provide more background information about the different Leishmania species you focus on. What are their hosts (you briefly mention lizards)? What is their distribution?

I also think you should provide more background information about assembling mitochondrial genomes from whole genome sequencing reads. Can you summarize past research that uses similar or different approaches? Are there examples outside of Leishmania? The end of your Introduction indicates that the pipeline development is the main objective of this study, so you should give a broader context for assembling mt genomes from Next Gen data.

2) Also regarding objectives, the last paragraph of the Introduction should more clearly state the objectives of this study and link more directly to your Discussion. The Discussion focuses almost exclusively on the phylogenetic utility of maxi and minicircles; the last paragraph in the intro should clearly state that this is one of your objectives, along with your objective of developing a pipeline for assembling the mt genomes.

3) You should include additional rationale for some of your Methods, For example, you describe why you performed PCR of L. major (Lines 177-180). I think this rationale should be mentioned in the Methods. Also, some sections in the Results should be moved to the Methods. For example, the methodological descriptions for additional proof of concept assemblies (Lines 251-259) is more appropriate for the Methods. In general, any methodological description should be in the Methods section.

4) Did you sequence individuals or pooled individuals of Leishmania? Also, was each Leishmania species sequenced on a single Illumina lane, or were they multiplexed? If they were multiplexed, I would be concerned about using unmapped reads to assemble the mt contigs, because barcode swapping can occur in up to 9% of reads on the HiSeq2000 machines.

5) I think you should add a section describing the broader implications of your pipeline. Could this approach be used in other systems that have variable mt genome structures?

6) The manuscript reads pretty well overall, but I think another round of editing for language and clarity would be helpful.

Some additional minor comments:

- An overview figure of the mt genome structures in trypanosomatids would be a helpful complement to the Introduction.
- Line 30: What do you mean by a “successful” group of parasites?
- Line 55: Do you mean the mitochondrial genome structure of Leishmania tarentolae is the best characterized among Leishmania? It currently reads like it is the best-characterized among eukaryotes.
- Line 63: Which six transcripts do not require editing?
- The Leish-ESP website is very useful, but are you also planning to submit the annotated maxi and minicricles to a long-term database (e.g., NCBI’s GenBank or ENA)?
- Why did you use Tamura-Nei models in your phylogenetic analyses? Did you test for optimal substitution models based on e.g. AIC?
- Please change the colors of the boxes in Figure 3. They are difficult to see.
- Figure 5: are the bootstrap values not shown all <50?
- Figure 6A: Some of those bootstrap values are very low. It might be worth briefly discussing this.

Reviewer 2 Report

This study by Camacho et al demonstrates that relatively short reads (125-bp) from Leishmania whole genome sequencing data can be used to completely assemble the conserved (coding) sequence of the mitochondrial maxicircle and, less reliably, at least part of the minicircle population. The authors further show that the maxicircle sequences can be used to produce robust phylogenetic trees that reflect known relationships at the trypanosomatid species and family level. Quality of data and presentation are usually of a high standard (but see detailed comments below). Most results are confirmatory rather than novel, but they are still very useful, and the study will be of considerable interest to the trypanosomatid research community. Some issues concerning inaccuracies, lack of clarify or missing data should be addressed, however.

Line 41. Some gRNAs are encoded in the maxicircle.

Line 43. Including more up-to-date references on RNA editing in trypanosomatids would be useful for readers of this manuscript.

Line 47. Many gRNAs in Leishmania are redundant (see Simpson et al. 2015, PMID 26204118), so loss of a minicircle class does not necessarily disrupt the editing cascade.

Line 51. It is actually not known if the two daughter minicircles are inserted at the antipodal sites of a kDNA network. It is only known that replicated minicircles are inserted at both sites.

Line 56. ’10,000 - 20,000 minicircles’. The source quoted states 5,000 - 10,000 minicircles.

Line 63. MURF5 has been identified as component uS3m of the mitoribosome (Ramrath et al. 2018, PMID 30213880). G3 and G4 have tentatively been identified as subunits of respiratory complex I (Duarte & Tomas 2014, PMID 24961227).

Lines 104-119. The authors state that they have developed an assembly pipeline, but the section in the methods is extremely short on detail. Please describe the process in sufficient detail (parameters used etc.) that it can be reproduced by others. Bespoke bioinformatics scripts need to be deposited in a public repository.

Line 113. What cut-off was applied in the BLAST search to identify contigs with sequence homology? As, for minicircles, the blocks of high conservation are short, care must be taken at this step, or genuine minicircle sequences could be lost. Other studies have relied on size and presence of CSB-3 for identification of minicircle candidates.

Line 183. For one Trypanosoma brucei strain, the complete maxicircle sequence including the variable region has been determined (Sloof et al. 1992, PMID 1336570).

Line 189. It is very surprising that the MURF5 gene could not be identified in any of the maxicircles assembled here. Did the authors try to BLAST translation products from all 6 ORFs against the MURF5 protein sequence?

Line 199-201. This text belongs to the legend for Figure 1, not to the main text.

Tables 1-3 could be combined to save space and facilitate comparison.

Line 213. Please use ‘pan-edited’ instead of ‘highly edited', as this is standard terminology.

Section 3.3. Please include a chart that shows the sizes of the assembled circular minicircles.

Line 265. What was the criterion for calling two minicircles ‘different’? Within a population, minicircles share different degrees of relatedness, as also reported here. Please report the cut-off that was used for calling two minicircles ‘the same’ (as in case of the reported minicircles distinguished by ‘polymorphisms’ below) vs. distinct.

Line 269. Minicircles sizes are species-dependent, but within a species heterogeneity has been reported to be limited. Is this different in the case of the minicircles identified here?

Line 285-289. Such a drastic reduction in complexity of the minicircle population in Leishmania would be unprecedented (PMID 26204118 should be cited here) but could readily be tested by comparing the number of sequencing reads with the CSB-3 sequence in the three sets. If the (normalised) number is comparable for L. braziliensis then the likely cause is a problem with assembly, not a lack of minicircles in the strain). Similarly, mapping back all reads to the assembled minicircles and comparing mapped vs. unmapped reads with CSB-3 would provide a measure for the efficiency of the assembly.

Line 306. Please indicate the second CSB-2 like sequence in Fig. 3C.

Line 326-333. Micro-heterogeneities or polymorphisms within a minicircle class have been reported before (Simpson et al., 2015).

Figure 5. Please report bootstrap values for all branches.

Figure 6. The legend mentions that maxicircle sequences for L. adleri and L. guyanensis were used in this analysis and had been generated as part of this study. Please amend the Methods and Results sections accordingly and provide database accession numbers (see below).

Lines 415 - 429. I find the arguments presented here in favour of using the maxicircle for phylogenetic analysis somewhat inconsistent. Phylogenetic analyses based on minicircles (their ref 46) and maxicircles both require NGS methodologies. A discussion of the pros and cons would be useful for the reader. Also, discussion of utility as marker sequence for sensitive detection vs. phylogenetic analysis could be distinguished more clearly.

All maxicircle and minicircle sequences assembled here must be deposited in public databases before publication. Please provide accession numbers in a revised version of the manuscript.

There are some problems with typographical or syntax errors. Please proofread carefully and revise where necessary.

Round 2

Reviewer 1 Report

I commend you on your thorough attention to my previous comments. I think the manuscript is greatly improved. 

My only previous point you could consider addressing is regarding the choice of substitution model for phylogenetic analysis. Personally, I would perform a formal model test based on your sequence data (e.g. with ModelTest or PartitionFinder). However, I doubt you would see major differences in your results, so it's probably OK to leave it as is. At the very least, please cite the previous studies that use the TN model as a rationale for why you used that model.

Author Response

We did not know the tools ModelTest and PartitionFinder. Now, we have read about PartitionFinder 2. Certainly, we must use it. It seems to be a user-friendly application. Nevertheless, we need to download, install the program, and, more importantly, to learn how it works.

Therefore, we took the alternative of citing previous studies that use the TN model and MEGA software. This approach was used by Kaufer et al (Ref. 54) and by Asato et al (Ref. 64). Moreover, it is recommended in a recently published methodological article dealing with Phylogenetic Studies in Leishmania. This article has been written by Katrin Kuhls and Isabel Mauricio, two well-recognized experts in taxonomy of kinetoplasts. This article corresponds to Ref. 65.

Thank you for this advice, and the comments in your previous report. You have taught us very relevant concepts. Thanks!